# "Why Drones for Ordinary People?" Digital Representations, Topic Clusters, and Techno-Nationalization of Drones on Zhihu

**Andrea Hamm [1,2],\* and Zihao Lin [3],\***

[1]  Weizenbaum Institute for the Networked Society, 10623 Berlin, Germany
[2]  Faculty of Electrical Engineering and Computer Science, Technische Universität Berlin, 1
     0587 Berlin, Germany
[3]  Department of Social Sciences, Humboldt-Universität zu Berlin, 10117 Berlin, Germany
\*   Correspondence: andrea.hamm@tu-berlin.de (A.H.); zihao375@foxmail.com (Z.L.)

**Abstract:** Unmanned and unwomaned aerial vehicles (UAV), or drones, are breaking and creating new boundaries of image-based communication. Using social network analysis and critical discourse analysis, we examine the 60 most popular question threads about drones on Zhihu, China's largest social question answering platform. We trace how controversial issues around these supposedly novel tech products are mediated, domesticated, visualized, or marginalized via digital representational technology. Supported by Zhihu's topic categorization algorithm, drone-related discussions form topic clusters. These topic clusters gain currency in the government-regulated cyberspace, where their meanings remain open to widely divergent interpretations and mediation by various agents. We find that the largest drone company DJI occupies a central and strongly interconnected position in the discussions. Drones are, moreover, represented as objects of consumption, technological advancement, national future, and uncertainty. At the same time, the sense-making process of drone-related discussions evokes emerging sets of narrative user identities with potential political effects. Users engage in digital representational technologies publicly and collectively to raise questions and represent their views on new technologies. Therefore, we argue that platforms like Zhihu are essential when studying views of the Chinese citizenry towards technological developments.

**Keywords:** drone; UAV; Zhihu; techno-nationalism; China; digital representational technology; digital citizenry; social network analysis; critical discourse analysis

## 1. Introduction

By today, China is the country of the largest global consumer drone producer. We see social changes parallel to the emergence of drone technologies and their marketizations in China within the last two decades. Unmanned and unwomaned aerial vehicles (UAV) or drones are breaking and creating new boundaries of image-based communication. Authoritarian state policy, global business-making, and civic engagement have various and sometimes contradicting interests towards drones. All three parties aim to leverage the technology for themselves. The state utilizes drones for social control and national identity-building, companies gain high profits from the drone consumer market and claim the ambition to keep technological world-leadership, and civic drone pilots become equipped to go beyond everyday boundaries and to craft the tech-enhanced views to larger audiences via social networking sites (SNS) such as Sina Weibo, WeChat, or Zhihu. At the same time, new boundaries appear to limit and regulate the non-state use of drones and their dissemination. During

these processes, drones constitute not merely material entities, but also objects of discourse raising clusters of controversial issues through which social changes are made viable.

To our knowledge, researchers have not yet empirically investigated how drones are shaping and shaped by the contemporary Chinese society. The birth, development and widespread embrace of drones are a part of China's broader dynamics of a giant digital leap which is characterized by the multitude of tech-savvy netizens on the one hand, and omnipresent state regulation on the other hand [1]. In this regard, social media platform proffers a window for capturing the changing perceptions about changing subjects. In 2018, for example, the largest Chinese social question answering site "Zhihu" has been sanctioned for "not censoring enough content" [2] when netizens have criticized the Chinese president Xi Jinping.

Asking "How are drone-related social changes understood and represented by users on Zhihu?", we carry out this study in a framework of e-participation mediated in a techno-nationalist environment. We argue that multiple representations of drones and what they mean to Chinese society can be understood by critically studying user-generated contents on SNS which we conceptualize here as digital representational technologies allowing to conclude user's understandings of technologies.

In this explorative, multi-method study, we first carry out social network analysis to receive an overview of the drone topics on Zhihu. Second, we conduct a critical discourse analysis to understand how drones are mediated by users as social objects that change Chinese society, and also the world. We understand the entanglement of the two technologies, SNS and drones, as essential for such development.

## 2. Context

### 2.1. Drones and the Chinese Digital Economy

In the last decade, the Chinese government and corporations have largely invested in the information communication and technology (ICT) sector to make China industrially and technologically independent from foreign companies, and at the same time, a global technological leader. Hong [3] identifies that the state and the elites failed to reshape social logics and redistribute power and resources, even though China's global impact has tremendously increased due to modernized infrastructures and the digital economy. Instead, the national technological development would be prioritized over balancing economic growth and social justice [3]. A recent example is the 9-9-6 working model, which is by now reported in global and Chinese media. The 9-9-6 formula describes the overtime workload in Chinese leading technology companies: Working from 9 a.m. to 9 p.m., six days a week. Initially, an organized group of tech workers published a blacklist of 150 major tech companies on GitHub; since then the issue is now intensively discussed regarding exploitative company practices and the social cost of technological leadership [4].

On this blacklist, one can find DJI Dajiang Innovation Tech Co Ltd., that is, the largest consumer drone manufacturer worldwide [5,6]. Allowing its customers to steer a drone by swiping on the smartphone screen or taking pictures with 4K resolution, the company has gained tremendous popularity among tech fans, aerial photographers, and creative video producers.

DJI presents itself as an innovative tech-leader company. In its public relations newsroom, articles show the current communication strategy of DJI company. On the one hand, these articles report positively on the company, describing software improvements, awards obtained, and the successful adaptation to new government drone regulations; on the other hand, the newsroom defenses heated scandals against DJI such as employee fraud and battery investigation [7].

DJI cares a lot about visibility and reputation among its supporters. All customers are nudged to download DJI Go, the official smartphone application, to access the newly purchased aerial devices. With a built-in SNS module, the app allows registered users to preview, edit, and share the drone footages. It creates user networks densely connected to the company and its services. Additionally, the company is present on major Chinese digital platforms and seeks active user interactions. On Sina Weibo, DJI has almost 990,000 followers and posted more than 8100 Weibos

(April 2019). On the social question answering (SQA) platform Zhihu, the DJI topic has more than 34,000 followers and contains more than 3000 questions (April 2019). Around the DJI drones, networks of interested netizens participating in multiple drone-related conversations have evolved during the past years.

## 2.2. Drones and State Authorities

Parallel to the booming drone consumption in the private sector, the People's Republic of China's state authorities have leveraged drones for public uses as tools of social control and law enforcement. More than one thousand public service drones have been introduced nationwide [8]. The media report that the public police applies drones equipped with object recognition, positioning, trajectory prediction for "combat missions under complex conditions" [9]. Against traffic violations, drones have been tested in the Sichuan region as recording tools for illegal driving in emergency lanes [10]. For surveillance practices, a dove-looking drone has been deployed by more than 30 military and government agencies in at least five provinces, mainly in areas considered by the government as "separatists' hotspots" [11]. The reports draw a thin line between the more effective drone-enhanced policing and the potential misuses of the technology by authorities.

State authorities also leverage drones as tools for their mass communication strategies. Drones create exceptional footages from an aerial perspective. In 2017, CCTV Documentary International Media Co. Ltd. has published the documentary "Areal China" picturing a compilation of Chinese landscapes recorded by drones [12]. The aerial perspective is aesthetically refreshing and pleasing, creating images of Chinese landscapes from the same transcending angle. We see such materials are being used for national identity-building strategies: The diverse landscapes are collected in one video content to represent one Chinese nation while implanting an impression of technological leadership and enhancement.

## 2.3. Drones and Activism

Drones can be regarded as new tools for the civil society making efforts to social change [13] and social benefits [14]. Activists used drones to deliver water, food, medicine, and media contents to North Korea [15,16]. In China, the photographer Wu Guoyong collected aerial footages of bicycle graveyards depicting several thousands of unused bicycles lying on abandoned industrial sites in mainland China to illustrate questionable business practices [17]. As such footage has been disseminated via SNS, they created a popular issue calling for collective attention and the displayed malpractices became evident and worthy of criticism.

The drone becomes a powerful tool of documentation for civic engagements [18]. During confrontational protests, journalists and citizens recorded events involving conflicting parties, notably during the Umbrella Movement in Hong Kong in 2014 [19,20]. Such footages capture the visible extent of the particular social tension and the on-site reactions of the police. In this way, drone pilots construct a seemingly elevated and distanced position in the actual events. Drone pilots can act as watchers, while the public authorities and protesters are being watched. This boundlessness of the airspace allows drone pilots under certain circumstances to act beyond the aviation regulations; by publishing such footages on SNS platforms, even censorship rules can be avoided [21].

## 3. Conceptual Framework

### 3.1. Beyond Being a Tool for Civic Engagement: Drones and Their Digital Representations

The understanding of the drone as a tool for social movements having transformative potential existing in the literature [14–16,18] is too simplistic and taken for granted in our view. In addition to this utopian approach, one can also find a dystopian view on the drone that makes people worried about their privacy and safety, increased surveillance, and psychological effects [22]. We suggest going beyond this dichotomic logic. For us, drones should be best understood as an open assemblage of various heterogeneous mediators in a Latourian sense. In this sense, drones are discursive objects that can be represented in many ways. Due to user' participation in online platforms, clusters of

controversial issues are raised through which social changes are made viable. Relating to Latour, we see drones as mediators that "transform, translate, distort, and modify the meaning or the elements they are supposed to carry" [23] (p. 39).

Further, we propose that the entanglement of the two technologies, SNS and drones, has the potential for further social change. As a growing sector of the digital economy, the drone industry itself is reliant on existing SNS platforms to cultivate its visibility and attractiveness. On SNS, users have the opportunity to generate contents and are psychologically motivated to do so [24]. The potential large range of disseminated contributions [25] and the ubiquitous attention-attracting visual contents [26] have changed how contents are spread among people.

In this manner, we suggest the understanding of SNS as variations of digital representational technology: Users represent themselves on digital platforms where the content is fully mediated via information technologies. This computer-mediated way of communicating shifts the practices of face-to-face exchange or other forms of assembly on non-digital places to fundamentally new ways of hyper-personal interaction [27]. The representation includes a self-representation and a representation of one's view of the world. The active practices of representations, relationship building, and knowledge sharing are crucial fabrics in the formation of virtual community [28]. The SNS and civil drone technologies provide the necessary infrastructural and informatics conditions [29] that enable the virtual community to stabilize and develop.

### 3.2. Techno-Nationalization of the Chinese Internet

Our study is conducted in the techno-nationalizing environment of the Chinese internet. In rapidly growing Asian countries, techno-nationalism has often been used to describe technology policies intending to gain technological independence from the West [30]. The two main aspects of techno-nationalism are (1) the belief that "technology is a crucial national asset in a highly competitive world" [30] (p. 913); and (2) the pragmatic mix of liberal and nationalistic policies to reach the national technological goals [30]. After the Chinese president Xi Jinping came into power in 2012, the China Dream as a "defining slogan" [31] (p. 7) became highly popular. It names the "great rejuvenation of the Chinese nation" which describes "the revival of the hard-working and hardship-stricken Chinese nation after the Opium War taking place about 170 years ago" [32]. Plantin and de Seta [33] recognize that techno-nationalist media regulations and a cyber-sovereignty agenda have created the gated "Chinese internet" dominated by the large Chinese technology companies. The abbreviation BAT names the leading company trio containing Baidu (search engine), Alibaba (e-commerce), and Tencent (messaging). Such techno-nationalistic shaping is especially visible in the WeChat model, a ubiquitous platform in Chinese daily life used for paying, messaging, organizing, streaming, gaming, etc., in this way WeChat became an increasingly critical infrastructure for Chinese people [33].

Following this conception, we refer to techno-nationalism in the sense of discourses or ideologies (rather than practicable policies). Tracing narratives of drones in relation to national sentiments on the Zhihu, we are interested in exploring how both drones and SNS can be domesticated as key objects of global technological leadership for Chinese policy, economy, and society.

In Western democracies, "social change efforts succeed or fail based on their ability to sway public opinion" [13] (p. 48). In China, it is difficult to measure the public opinion from state-controlled media outlets. Scholars identified a fragmented public sphere on Chinese digital platforms that reflects multiple tensions between citizen participation, journalism, and the government [34]. On Sina Weibo for example, politically controversial issues emerge and can become widely distributed by elite accounts, even though the platform is fundamentally regulated by the state [35]. The drone use of Chinese state authorities is commented, endorsed and questioned by SNS users: Weibo users appreciate that delinquents will be identified by using traffic monitoring drones, at the same time, a traffic policeman is asking if this would be the beginning of "tech-enhanced policing" [10].

We must note that users on Chinese digital platforms act within a heavily regulated environment where state actors can act as both judges and players [35]. On Zhihu, Weibo, and other platforms, users have to register via a real-name registration and indicate a valid phone number. Different from

the rather anonymous Western platforms, Chinese netizens are directly identifiable by platform providers and hence by state authorities. At the same time, many social and political issues cannot be addressed publicly since the Chinese state exerts a thorough censorship policy [36] and fine-tuned filter algorithms based on sensitive keywords [37]. The risk of being identified and the Chinese censorship policy are two factors that can be assumed to cause a high degree of self-censorship or chilling effect [37].

On social question answering (SQA) platforms, users ask informational and conversational questions. Informational questions seek fact-oriented answers, while conversational questions intend to stimulate a discussion to receive people's attitudes or acts of self-expression [38]. Zhihu is the largest SQA platform in China. Managed by its own corporation, it does not belong to the big three online companies BAT. Different from news media that publish articles due to current events, on Zhihu, it might be mainly the need for orientation that leads users to the publication of questions. At the same time, users can present themselves as experts of a certain field and receive rewards for great contributions. In sum, Zhihu provides a platform for users that seek knowledge, social orientation, or (self-) representation.

## 4. Research Design

### 4.1. Research Question

The overarching research question of this study is: How are drone-related social changes understood and represented by users on Zhihu? We understand both drones and SQA platforms as mediators for social change in the Chinese society. Drones as a tool or product are leveraged by all societal groups. Since the different interests toward drones can lead to tensions between the traditional powers of the state and the civil society, in turn, different discourses circulating via social media provide us a window to peep into and analyze the divergent opinions, experiences, emotions, and conflicting interests between concerned groups participating actively in online spheres.

### 4.2. Pre-Exploration of Zhihu Platform Characteristics

The SQA platform Zhihu is organized on a topic basis and does not limit the number of allowed characters in a user contribution. It represents a large knowledge resource for registered users but also for non-registered netizens who can access the platform without log-in. Registered users can contribute to topics by posing a new question, giving an answer to an existing question or commenting on it; they continuously extend the accessible resources and shape the visibility of contents on the platform by endorsing the user contributions.

Our pre-exploration of Zhihu reveals how its technological and algorithmic settings will nudge netizens to use the platform in very specific manners. To pose a question, one needs first to log in; the successfully registered account must be linked to a cellphone number (not necessarily a Chinese domestic cellphone) in the state requirement of real-name identification [39]. Having a verified account, one can click the button of "post a question." A pop-up window will appear with a text field ("write down your question; accuracy can help you get answers more easily") and there is an option of "post the question anonymously." While typing in the text field, a dropdown list will appear, showing some relevant questions and respective numbers of their questions that already exist on Zhihu ("your question might already have been answered"). After having finished drafting the particular question (our example is "Is the regulatory framework for drones stricter in China compared with other countries?"), another text field states; "Type in detailed descriptions such as the backgrounds or conditions of the question (optional)." One can find the option to edit the fonts, inserting pictures or videos. Below in the description field, two topic categories automatically appeared ("UAV" and "Guojia/State"). These are extracted from the keywords of the question based on the platform algorithms. In another text field, one can add up to five topics manually. Then, when typing topics manually, the dropdown list appears again with already existing topics that might match the manually given words [40].

After having posted the question, one automatically receives a follower of the posed question. For each topic, there is the option "Invite to answer," which promotes the interactivity between users. When pressing the invitation button, a pop-up window appears with a list of automatically suggested users. There are three dropdown options for a given question: (1) Canceling the anonymous identity (only accessible to the question poser); (2) deleting the question (only accessible to the question poser); (3) viewing the question log (accessible to every viewer). The log records all the versions and editions of the question and its description and its topics. The public editing system reflects how Zhihu service providers highlight the publicness of the questions. They state in their platform instructions ("Knowledge Guide") that the questions collected on Zhihu are "public resources, just like the entries on Wikipedia" [41].

### 4.3. Data Collection and Methods

Our data were collected between 14–28 February 2019. The research material contains the 60 question threads with the highest number of endorsements in the topic "Drone/UAV" on Zhihu. In our analysis, we only consider such questions, answers, and comments that are the most visible in the platform. The visibility of content is indicated by the number of endorsements ("likes") which is assigned by users themselves. We decided to choose the visibility approach for two reasons: (1) The ranking of question/answers has been voted by active user participation, and (2) new users or passive readers will be exposed to this ranking of questions/answers when they search by relevant topic keywords.

We first provide a general mapping of our results based on social network analysis (SNA) and then further the interpretative findings by critical discourse analysis (CDA). The textual and the visual corpus on Zhihu is very complex both in terms of the platform's multi-media forms encompassing images and videos, and the various narrative strategies adopted by its tech-savvy users. Thus, we take an explorative, qualitative way of conducting CDA in a social media environment as demanded by Bouvier and Machin [42]. Unless otherwise noted, the translation of research materials from Chinese to English is provided by the authors of this study.

We extract the various representations of drones expressed on Zhihu to make further conclusions on drone-related social changes. According to Van Dijk, three forms of social representations are relevant to understand a discourse: Knowledge (personal, group, cultural), attitudes (cognitive and emotional), and ideologies; as discourses emerge and evolve within societies, they can only be understood by considering the respective social situation, actions, actor and societal structures [43] (p. 21).

Attitudes are schema-like organizations of general opinions and beliefs toward a social object that can be a thing, a person, a group, an event, or an issue (or something similar) which is relevant in social interaction for social members [44]. People have attitudes to social objects such as nuclear energy or birth control, but not about apples since they "are not involved in a complex system of social interactions" [44] (p. 40). Attitudes, comprising a system of opinions, are themselves organized in a system of ideologies: An ideology comprises a system of attitudes; and ideologies are the "fundamental system underlying social interaction" [44] (p. 40). We see the drone as a social object for the society on which people have different opinions and beliefs that can be crystallized into knowledge, attitudes, and ideologies towards drones. We aim to reveal these representations from the Zhihu discussions to describe the ongoing drone-related social changes.

## 5. Findings

### 5.1. Exploring Topic Clusters Around Drone by SNA

On Zhihu, every question thread is categorized by topics. These topics are algorithmically recommended by the platform and need to be manually finalized by the question author at the time of question creation. In this way, a question does most likely not only appear in one topic but in several. For example, the question "What kind of new experience is taking a selfie with a drone?" is categorized within four topics: Photography, selfie, drone/UAV, and aerial photography. By being

visible in several topic threads, drone-related discussions unfold in what we term "topic clusters". Mapping such topic clusters provides insights into the different perspectives under which drones are represented by Zhihu users.

Social network analysis (SNA) allows visualizing relationships and interconnections of related items [45]. We use SNA to map the drone topic as a discursive object with many interconnections to related topics on the Zhihu platform. The generation of the SNA graph reveals the inclusivity as well as the fragmentation of the drones as a multifaceted discourse. We can identify which topics have the strongest linkages to the drone topic and how other topics are interconnected with each other, e.g., if groups of topics can be found.

Figure 1 shows the SNA graph based on the topics of the 60 most visible Drone/UAV questions on Zhihu. Each node represents a topic assigned to a question. Each question is organized under up to five topics selected by the question author. The graph shows a total of 100 nodes, with the Drone/UAV topic appearing in the center and representing the starting point of the network. The 99 topics are interlinked around the Drone/UAV node. To generate the graph, we used the open-source visualization software Gephi [46] and the integrated algorithm Force Atlas, which is recommended for SNA [47].

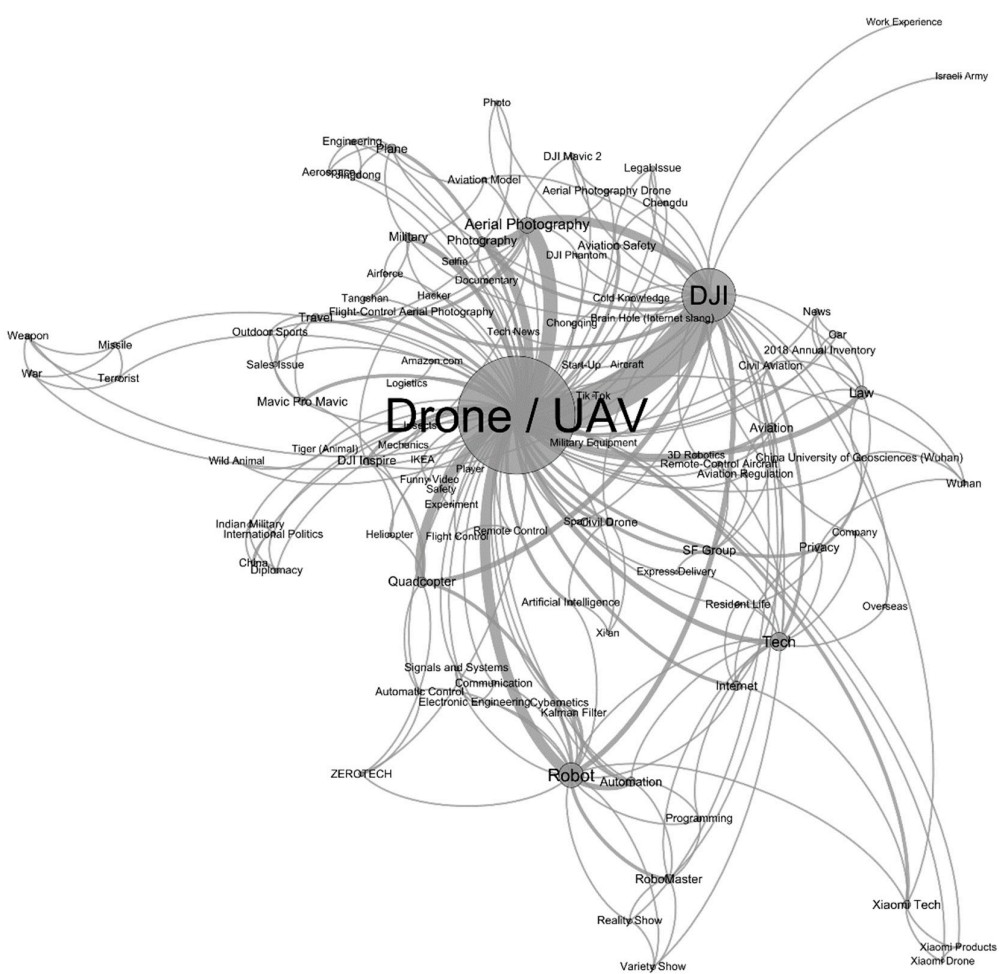

**Figure 1.** Topic cluster graph around the topic Drone/UAV on Zhihu; generated by Gephi software using the Force Atlas algorithm. The 60 questions ranked by their most endorsed answers have been collected in February 2019, see Appendix A.

The network graph shows that drone-related topics on Zhihu are either clustered in groups or scattered over the graph. The largest nodes after drone/UAV are DJI Dajiang Innovation, robot, tech, aerial photography (by decreasing size). The node size implies that these topics have respectively

become a coherent sub-theme that can be differentiated from the others. Around these dominating sub-topics, we identify the four topic clusters that are partly overlapping each other:

- DJI Dajiang Innovation is connected to, e.g., work experience, Israel Army, Chengdu, aerial photography, aviation model, aviation safety, tik tok;
- Aerial photography is connected to, e.g., selfie, documentary, photo, photography, DJI Mavic 2, travel, outdoor sports, aviation model;
- Robot is connected to, e.g., electronic engineering, automation, artificial intelligence, quadcopter, experiment, flight control, remote control
- Tech is connected to, e.g., Internet, overseas, resident life, privacy

Not every topic that we assume to be interesting (based on the literature review) is clearly clustered in the graph. For example, law is a larger node in right part of the network, whereas similar topics like safety, aviation safety, aviation regulation, legal issue are scattered over the graph. This implies that Zhihu users do refer to drone-related legal issues, but in a less consistent manner. Additionally, we find topics of international policies and politics that deal with military drones and not with the consumer drone, scattered over the network graph. Related keywords to this part of the discourse are weapon, missile, terrorist, war; military, Airforce, etc.

Studying the distribution of the number of views related to the number of endorsements, we found some remarkable outliers (see Figure 2). The question "What are the experiences of working at DJI?" (qID 23) has the highest number of views in the sample, but a rather low number of endorsements. Considering the comments of this question (see Section 5.2.3), we assume there might be a chilling effect that prevents users from endorsing polemical answers of qID 23. In contrast, commercial questions, e.g., "As a user, what do you expect from the Xiaomi drone?" (qID 1) and "How to evaluate Mavic, a new product released by Dajiang Innovation?" (qID 2), attract high numbers of endorsements. This popularity might be related to the public relations activities of the companies DJI and Xiaomi ongoing on Zhihu, which lead to the comparably high numbers of endorsement.

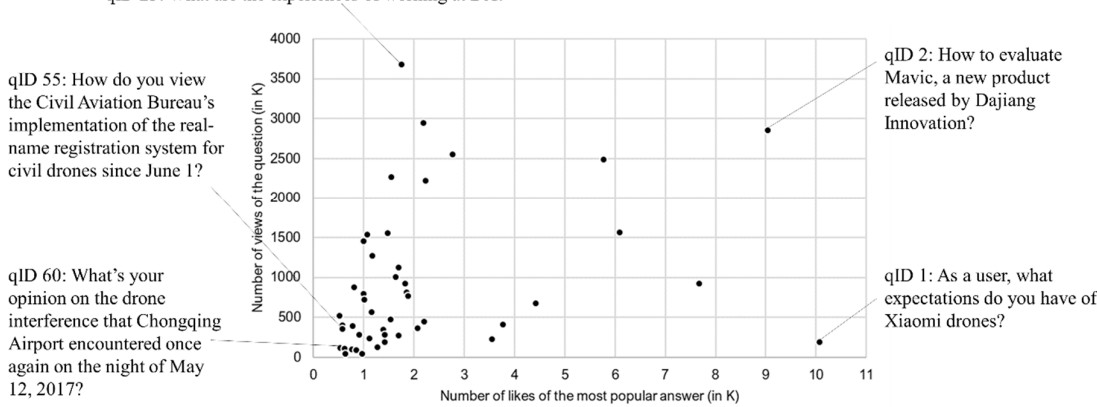

**Figure 2.** Distribution of the number of views of a question related to the number of likes of the most popular answer. The latter defines the question ranking in the Drone/UAV topic on Zhihu. Some questions are highly viewed, but their answers have a disproportionately low number of likes, e.g., qID 23. Other questions show rather high numbers of endorsement, e.g., qID 1 and qID 2.

Compared to these outliers, the largest part of the questions has fewer numbers of views and endorsements, some of them being too specific and thus less recognized by large numbers of users, e.g., "How is the modern advanced control theory advanced?", "How do you view the situation where the advanced control system is actually not that stable even though the PID controller has almost 90% share?" (qID 48) or "How to explain the Kalman filter as much as possible and as detailed as possible?" (qID 50). Other relatively less viewed and liked questions ask about sensitive incidents

and regulations. Here it might again be a certain chilling effect keeping some questions less visible, for example: "How do you view the Civil Aviation Bureau's implementation of the real-name registration system for civil drones since June 1?" (qID 55) or "What's your opinion on the drone interference that Chongqing Airport encountered once again on the night of 12 May 2017?" (qID 60).

*5.2. Identifying Drone Representations by Critical Discourse Analysis (CDA)*

For each of the topic clusters named above (DJI Dajiang innovation, aerial photography, robot, tech), we respectively include the three most visible answer threads or articles for critical discourse analysis. Users can write comments on the question, offer answers, or add comments on the given answers or articles. We will combine these contributions in the analysis as a whole. Attitudes are embedded in the discussion of the themes and hence will be analyzed together with the thematic dimensions.

We find that users apply multiple rhetorical and illustrative techniques to decorate their questions and answers. Zhihu contains rewarding functionalities for users succeeding to engage other users to contribute to their question. Therefore, we interpret this meticulous style of content contribution as a behavior seeking to attract other users while creating a community persona of the self.

In general, we reveal at least three different representations of drones expressed by engaged Zhihu users. We analyze in the following the users' understanding of drones and the users' representations towards the drone in a Latourian sense as a mediator of social change.

5.2.1. What Qualifies a Drone?

Drones are characterized as a new kind of camera, enabling an aerial view of the world. Users contributing to this part of the discourse are hobbyists or (semi-)professionals leveraging drones as image and video recording gadgets. As tools for creative digital self-representation, it is mainly the consumer perspective that is strongly promoted by DJI Innovation Tech prevailing here. Users post their photos and videos shot by drones and represent themselves as enthusiastic drone pilots. The interdependence of SNS and drones becomes obvious since users seek gratifications for their contributions. A user answers the question "What kind of new experiences is it to take selfies by a drone?" (qID 13) by posting holiday drone footages and representing himself as traveling happily with his girlfriend. The most endorsed comment to this answer praises the video and makes a compliment to the girlfriend:

"During the summer holiday, I traveled with my girlfriend to Xiamen. It feels so happy to see the world from a different perspective with a loved person. (a video inserted here) […]" (qID 13)

"She is stunningly beautiful; I want a girlfriend like her" (qID 13)

The texts here become rather personal or even intimate, showing personal travel experiences and visual materials of the girlfriend to the Zhihu community. Drone-enhanced images and videos become important parts of the question and the answer to illustrate views, experiences, and artistic skills. For example, a user posts a photo probably of himself sitting alone on a mountain ledge with a colorful sky and vast mountain scenery in the background. Such a photo could not have been taken by a handheld camera. It appears that the self-representation of the users becomes much more important in this part of the discourse than the objective representation of drones themselves. Users create a certain persona of themselves as artistically skilled and probably seek gratification.

We find that not all Zhihu users have the same understanding of drones. Some users make a strong point about drones as objects of research and development that apply complex computing technology. They do not see consumer drones as drones due to a lack of technical functionalities. Within the question threads, "How big is the gap between Xiaomi Drones and DJI Drones?" (qID 4; list of all qIDs can be found in Appendix A) and "How to perceive the practices of SF Groups to start using large drones for express delivery?" (qID 20), users elaborate on what a *rotor aircraft* must fulfill to be called a drone:

"[…] a qualified multi-rotor aircraft must similarly obtain the intelligent navigation system consisting of inertial measurement components, GPS, visual odometer, obstacle avoidance system, barometer and ultrasonic sensors to be called an unmanned aerial vehicle. A multi-rotor aircraft is not a toy. […]" (qID 4)

"The reason why I don't like to call these things UAV is that I think one necessary qualification of UAVs is the anti-interference capacity. Ordinary quadrotors use 2.4 G for communication, which is so easy to fall prey to interference." (qID20)

Many users present themselves as meticulous technical experts or professionals by listing technical details and knowledge to illustrate that a drone or UAV requires sophisticated technical features distinguishing it from a simple aviation model. Such self-representation is underlined by categorical and determined statements, e.g., that a drone "is not a toy." Another user separates himself from "ordinary people" and questions if consumer drones are the right tools to execute various tasks; he sees "high-quality drones" as a more capable tool and clearly distinguishes those from consumer drones:

"For ordinary people, a low-end consumer drone is only possible as a toy. If one wants to prove that a drone is not a toy, then what else can a person do with a cheap drone other than using it for entertainment? I recognize that a high-quality drone can make contributions in the fields of aerial photography, agriculture, plant reservation, express delivery, disaster rescue […], and can be legitimately called an 'instrument.' But are Xiaomi and DJI low-end drones really qualified to be called 'instrument'?" (qID 4)

The answer thread of qID 4 is fully dedicated to technical details showing a deep knowledge on the functionality of the technology provided in long texts. It appears that due to the sweeping tech explanations, contributions of less tech-savvy users might be sidelined or receive at least less visibility. The high numbers of endorsement of answers focused on drone technicalities imply that Zhihu users highly appreciate and encourage the exhibition of engineering knowledge and technical competence.

In sum, the discourse here is characterized by self-claimed experts' knowledge and definitory discussions of the notion of the drone. Consumer drones are apparently not understood as drones by all users and pose subject of debates. Zhihu users endorse such insights and hence reinforce the visibility of such high-technical representations of drones.

### 5.2.2. Drones as Uneasy Flying Objects

The drone discussions reveal several traces of uncertainty. In the question "Where does one need to pay attention to while traveling with a drone for aerial photography?" (qID 31), this uncertainty is related to the baggage regulations of airlines, pilot certifications, and possible no-fly-zones at travel destinations. Due to the novelty of the technology, civic aviation regulations can change quickly, and the joyful experiences of drone enthusiasts may be reduced. A drone traveler expresses his lack of knowledge regarding the current regulations and recommends informing oneself:

"Not sure whether it's allowed to fly in Gulangyu now. Fellows here, you'd better learn the government regulations first before deciding whether or not you want to travel with your planes" (qID 13)

Here the state authorities become visible as an indirect source of uncertainty. State actors are not appearing as Zhihu users themselves, but the control and logics of regulation appear within contributions of other users.

Another representation of drones appears in terms of a consumer critique. In the articles "It only takes one after-sales service to make you transform from a DJI fan to an anti-fan" (qID 22) and "Why I wouldn't buy DJI drones again?" (qID 14), users explain their poor experiences with DJI customer supports. Such negative experiences stand in contrast to the happy consumer experiences that are narrated by DJI public relations or the hobbyist drone pilots. Zhihu allows customers to draw a broader image of drones as consumer products: Not only distributing approval of drone, but also criticism.

The appearance of drones in daily life makes some people concerned. These people are being watched by drones, but not using them by themselves. They are concerned by drones used by other people. The question "A drone is always flying around in front of my window; how to deal with it?" (qID 30) is posed together with a photo showing the drone in front of the building. This question relates to a privacy violation and noise pollution. The most popular answer is first mentioning that the drone is entering the "aerial territory" of the questioner, but then this tech-savvy user sophistically shifts this initially open question relating to potentially conflicting issues to his own representations of drones focusing on nothing else but technical details. He uses this answer to represent himself as an expert of the field and downplay alternative perspectives for example on privacy or noise pollution:

> "Very simple. First you can hang a warning on the window, say for example 'You have entered my aerial territory; if you don't leave within xx minutes, I will take necessary reactions.' Blah, blah, blah. Then, if this quadrotor (I really don't like to call these things drones) still doesn't leave, some special measures can be taken. […]" (the author uses the next paragraph to offer instructions on how to attack a drone with 2.4 G signals) (qID 30)

This "Blah blah blah" implies an attitude of irony and contempt towards what has been written previously. It entertains the possibility of protest by hanging a warning and provides a transition to the author's main argument that any other measure one could take against the drone would be in vain, unless it is signal interference. So, the original question referring to privacy concerns is apparently marginalized.

Uncertainty can also be found within the question "How to perceive the practices of SF Groups to start using large drones for express delivery?" (qID 20). The phrasing "How to perceive" is formulated in a sensitive way in order to not seem biased toward being too optimistic or too pessimistic about drones. The most popular answer is sarcastic and nonserious, however, it shows uncertainty and unease regarding future drone use by creating a reference to military bomb deliveries. A user on Zhihu answers the question by posting these official SF photos and stating: "Such a drone, such a container, you said that you are sending a courier, who would believe that!!!". Indeed, the photos show similarities to military airplanes and equipment.

A popular comment of this question is: "Two of China's leading express delivery enterprises, Dongfeng, and Shunfeng (SF)." Dongfeng is the name of a Chinese missile series, whereas Shunfeng is a logistics and delivery company. Such jokes or ironic comments probably contain the slight critique toward the use of initially military technologies for public services, or the uncertainty towards the malleability or transformative nature of the drone technology, in light of its obvious dual use in both military and civilian settings.

Underlying these contents, a critical view towards drones is present. Emerging drones are not directly criticized, but neither described as desirable technological innovations for China. Rather the drone's potential threat limiting one's personal freedoms occurs in an underlying message. Visual contents show drones or fearful consequences of drones, but almost no contents made by drones. For example, users post a photo showing a young child bodily injured in the face. The photo description makes clear that this accident is caused by a crash with a drone. Another user is doing experiments with a drone: Under the catchy title of the question "What is the killing power of drone rotors?" (qID 21), photos show how a drone is flying into a banana hung up by a thread, a second photos pictures deep cuts caused by the drone's rotors into the fruit allowing to estimate how harmful drone accidents can be. Representations of drones as an object of uncertainty encompass the domains of safety, lack of regulatory knowledge, and privacy. Such illustrations cannot be found in official media or outlets and therefore show a new perspective provided by questioners of drones.

5.2.3. A Successful Business, or Rather Not?

In the Zhihu discussions, we find that various users think of drones through a national lens. The question "Is DJI famous abroad?", "How do foreigners evaluate it?" (qID 43) is answered complexly

by setting DJI products in an historical context of weapons and relating it to the "China Dream," i.e., to become independent from foreign technologies and industries:

"During the first world war, we didn't catch the opportunity to become arsenal dealers. During the second world war, we became the waste treatment factories of arsenal dealers. All arsenal dealers took the greatest pride in selling weapons to both sides of the war. Now, regardless of ISIS or middle east tycoons, Lighthouse [US, A/N] marine corps or Mujahid in black ropes; they all use DJI. There are no more comments necessary for this. The China dream is realized" (qID 43).

At the same time, the remark "no more comments necessary" implies a certain irony and potentially critical view on the export of drones and maybe on the "China Dream" itself. The relevance of China for the drone market is also mentioned in a curious way around the question "What is the current market status of small civil drones and their future prospects?" (qID 29):

"Actually there are many companies working in this field, for example, AR. DRONE from Parrot, Phantom from DJI (which is created in China!)" (qID 29).

The exclamation mark emphasizes that the Phantom drone is not just made but also created in China, signaling the country's processes of innovations and technological advancements. By writing so, a sort of national pride for this fact is attached. Nevertheless, the user realizes that this remark does not so much belong to the question thread and puts it in brackets.

Drones are not only discussed as material entities but as a context topic. The innovation and popularity of drones allow questioning the working conditions in the booming Chinese digital economy. With more than 3.5 million views, the question thread "What are the experiences of working in DJI?" (qID 23) has the most views compared to all other questions in the Drone/UAV topic on Zhihu (see Figure 2). The question is described by:

"Rumor has it that DJI Innovation Tech has an arbitrary administration and the overtime workload is a serious issue. Don't know whether it is true or false." (qID 23)

The formulations "rumor" and "don't know whether" mitigate the criticisms addressed to DJI company: "Arbitrary administration" and "overtime workload" as "serious issues". To answer this question, actors from DJI are using national sentiments to frame drones. In their logic, the success of the company is connected to their national pride. At the same time, the success of single individuals (working at DJI) is connected to the DJI company. The most popular answer to this question is drawing an inseparable connection of success: The individual success that is linked to the company's success, which is moreover linked to the country's success:

"I've always thought of working for DJI as an honor. What we are doing is cultivating and leading a brand-new field of digital consumer goods. The innovativeness of Phantom in light of smart flight is similar to iPhone in smart phones. No other Chinese companies could create history like us, and merely that makes me proud from deep inside my heart." (qID 23)

"Successfulness" as a concept needs a certain object that it is linked to. At the same time, success is a personal projection. In the same answer, the user explains that it would not be the company's fault that people do overtime work, but the global competition:

"This flexible work and rest system were developed by countless overtime dogs (including me) […] I am doing the technical work I like, it is as important for me as breathing and eating. […] DJI has never forced anyone to work overtime, and it is not required to get a good job evaluation. But people who work in technology know that products cannot be innovated within the old nine-to-five logic. Who can make a breakthrough technology by working nine-to-five? If you want to be better than others, you have to work throughout the night, and you forget about everything around you." (qID 23)

The statement constructs a common knowledge, insisting that overtime is essential for the technological sector to remain competitive (or leading) and would not pose a problem for him as a

worker. By relating to feelings of national pride and general success, this answer sidelines criticism of overwork and declares overwork as indispensable and something to be "honored".

This thread on Zhihu shows that in the context of drones, one can ask questions about DJI as a working place and the question receives the highest number of views, i.e., more than 3.75 million. To answer the question, users elaborate on controversial issues such as overtime, welfare, and a stressful working culture. In contrast to the quote above, other users imply that DJI is not sharing its profits fairly with its employees:

> "Now DJI has developed into such a large scale, but still treats itself as a startup. This is clearly milking its employees, especially the young employees." (qID 23)

The sensitiveness of this question can be estimated as rather high, especially when considering the following user comments:

> "Just shared it with my classmates and found that many answers were deleted... Dajiang's public relations is amazing..."

> "DJI deleted a lot of answers."

> "With so many anonymous and deleted answers, the answer to this question is already coming out." (all quotes from qID 23)

## 6. Discussions

In the SNA visualization, we find that the strong linkages of the drone topic to DJI and aerial photography imply a prevailing consumer perspective on drones as tools for aerial photography mediated by the public relations of the largest drone producer DJI. The CDA however revealed that such consumerist view on the drone technology is only one among several representations of drones:

- As an object of consumption and technological advancement
- As an object of uncertainty
- As an object of national future

Our findings show that drones are actively domesticated by users via a national lens. Several answers and comments in our material corpus show a reference to the Chinese national development, global market leadership, or even weapon technology advancement. At the same time, the SQA technology engenders a space of attitudes to its users who are contributing in an intimate way to the virtual community. Drones open a discursive space for socially engaged people to make sense of the social changes that are unfolding. The changes encompass the emerging digital economies, the 9-9-6 work culture, new market regulation, consumerism, The China Dream, and technological breakthroughs.

In our study, we go beyond both the utopian and dystopian approaches to drone technology by taking a more nuanced perspective on how drones as a discursive object become socially constructed by multiple agents. The potential for social change of drones entangled with the SQA technology does not appear as an activist photographer's tool to shape an above view of the world or a protesters camera to monitor human rights protest or as an object of fear and human rights violation. Instead, the drone becomes a mediator for people to interact with and negotiate the more fundamental everyday changes. The appearance of jokes and irony (drone delivery), moments of uncertainty (how to deal with it, drone regulations) as well as the portrayal of private stories and images (DJI working conditions, drone traveling) show great open-mindedness and the self-representation enthusiasm of the users. We summarize drone-related self-representations that emerge from the Zhihu discussions as follows: Proud technical experts, happy and angry consumers, concerned citizens, honored entrepreneurs, and stressed laborers.

The Chinese state and authorities are represented indirectly on Zhihu and in the drone discourse. They appear as a judge that is defining the whole platform infrastructure in terms of community rules and making laws and regulations on drones, and rather not as a "player" [35]. However, coming back to the context of this study (Section 2), we find that the usage of drones for

public sectors, i.e., public surveillance, law enforcement, combat missions, is less debated on Zhihu than assumed. We find only a few traces of nationalist jokes related to military drones; besides that, there is a virtual absence of representation of "public drone uses" (as opposed to privately owned consumer drones).

With regard to the recently 9-9-6 working culture that is currently debated in media worldwide, our study displays another current relevance. We find the most viewed question ("What is the experience of working at DJI?" qID 23) is related to the working conditions at DJI company. The most popular answer elaborates on overtime, welfare, and working culture at DJI while claiming that such conditions would be suitable in the tech sector. This question on Zhihu even dates back to the year 2015, and the most popular answer was re-edited in 2017. The recently emerging public discussion on the extractive working culture in Chinese tech companies could hence already have been found on Zhihu for four years, though masked by the discourse of development aspirations and national pride. But even with such strong nationalist imprint, the sentiments of disappointment, stress, and even disenchantment of the life as "overtime dogs" get exchanged and disseminated via Zhihu. This reminds us to view drones not as portrayed by global media merely in the frame of readily packaged consumer products, but rather as objects of overworked people's labor, of people who are sharing their pride and discontent online just like their customers.

For more than 20 years, virtual communities are a rapidly growing phenomenon and are becoming increasingly important to our societies [27–29]. In China, this growth has been mirrored by scientific innovations: Major advances have been made with regard to technology, software development, and knowledge infrastructures. Drones appear as new tools in people's daily life together with the deliberative platform conversations around it. However, research to date often overlooks the affective and discursive ties that give rise to virtual communities. We argue that research that merely relies on the analysis of large-scale "big data" cannot serve to provide fine-grained details about meanings, attitudes, or opinions of virtual communities towards social changes or technologies. Quantified data of emerging discussion topics online alone do not provide enough context information to receive insights on people. Likewise, an uncarefully selected usually smaller corpus of online materials for discourse analyses may not be meaningful on a societal level.

We therefore propose to combine social network analysis (SNA) and critical discourse analysis (CDA) in order to reach a deeper understanding of the virtual community formation. We use the results of SNA to extract meaningful topic clusters which then serve as an access criterion to the textual materials for the CDA. To incorporate the essential materials for analysis, we choose the visibility of the material in the platform indicated by the number of endorsements ("likes") as a second criterion. In this way, the corpus is legitimated by a reasonable selection of relevant topics (by SNA) and the confirmed relevance of the individual post to a larger audience (by the number of likes).

Such carefully extracted data from an ongoing study of the emerging virtual community represents a pragmatic combination of quantitative and qualitative research methods for data collection. It allows complementing relational SNA information with interpretive findings from CDA. To correctly interpret the range and significance of such results, it is further essential to pre-explore the respective platform characteristics and societal contexts as we did here in the sections 2 to 4.

## 7. Conclusions

We hope to have illustrated by this explorative study that examining the processes of the self- and social representations in the face of a rapidly developing economy is equally important as examining the development itself. Users engage in digital representational technologies publicly and collectively to raise questions and represent their views on new technologies. The technology-related uncertain social changes become a source and motivation for users to participate in Zhihu in order to navigate themselves socially and collectively. Therefore, we argue that social networking sites like Zhihu are essential sources when studying views of the Chinese citizenry towards technological developments that cannot be discussed freely in usual Chinese media outlets.

The combination of results from social network analysis (SNA) and critical discourse analysis (CDA) yields insights into the topology as well as the content of a virtual community. By combining

these two methods, we bridge methodological shortcomings of both methods: SNA usually lacking meaningful context and CDA being highly time-intensive and hence restrictive towards the total amount of analyzable material. SNA provides data about the network and the relations among its actors, whereas CDA allocates meaning and sense-making within the network. With this study, we wish to substantially contribute to the methodological literature on virtual communities, in particular, to scholars who are interested in understanding the transformative power of social computing, computer-mediated communication, and community informatics. Such a mixed-method approach can offer grounded insights based on both the utilization of big data and the appreciation of thick data.

**Author Contributions:** Both authors equally contributed to the creation of this paper.

**Funding:** This work has been funded by the Federal Ministry of Education and Research of Germany (BMBF) under grant no. 16DII111 ("Deutsches Internet-Institut").

**Acknowledgments:** We thank the chairs of the 2019 ICA Preconference Digital Asia for their encouraging feedback to this study and especially the support of Muneo Kaigo.

**Conflicts of Interest:** The authors declare no conflicts of interest.

## Appendix A

**Table 1.** The 60 questions[1] with the most endorsed answer in the Drone/UAV topic on Zhihu.

| qID | Weblink | Question |
|---|---|---|
| 1 | https://www.zhihu.com/question/46540988/answer/102709473 | As a user, what expectations do you have of Xiaomi drones? |
| 2 | https://www.zhihu.com/question/277858902/answer/406016959 | How to view the incident where the son of an engineer from Mercedes Benz bruised a small child while playing with a drone, and the father of the responsible party claims to be unable to compensate. |
| 3 | https://www.zhihu.com/question/50964912/answer/123992298 | How to evaluate Mavic, a new product released by Dajiang Innovation? |
| 4 | https://www.zhihu.com/question/46752693/answer/103624548 | How big is the gap between Xiaomi Drones and DJI Drones? |
| 5 | https://www.zhihu.com/question/52237122/answer/191751196 | What kinds of things can you imagine that drones can do? |
| 6 | https://www.zhihu.com/question/38274006/answer/77993903 | Why drones for ordinary people? |
| 7 | https://www.zhihu.com/question/27039505/answer/39691301 | What are the new ways to play with the drone now? |
| 8 | https://www.zhihu.com/question/33279324/answer/56191019 | Why is there no drone self-destructive attack weapon? Have the terrorists used it? |
| 9 | https://www.zhihu.com/question/281760029/answer/423098332 | How to evaluate JD. com's plan to start the project of building an ultra-heavy drone with a weight of 40–60 tons and a flight range of over 6000 km? |
| 10 | https://zhuanlan.zhihu.com/p/35405080 | How did you shoot the Tik-Tok viral videos of flying high by mobile phone? |
| 11 | https://www.zhihu.com/question/309494263/answer/576790057 | How do you view the DJI Dajiang internal anti-corruption initiative that caused the loss of 1 billion Yuan and 16 people were handed over to the judicial treatment? |
| 12 | https://www.zhihu.com/question/58685856/answer/161852090 | How to view the incident on 17th and 18th April where drones caused the forced landing of civil aviation in Chengdu? |
| 13 | https://www.zhihu.com/question/52276436/answer/535363298 | What kind of new experiences is it to take selfies by a drone? |
| 14 | https://zhuanlan.zhihu.com/p/36749515 | Why I wouldn't buy DJI drones again? |

| 15 | https://www.zhihu.com/question/484 73315/answer/111489820 | Wishing to buy a drone and use it when traveling outdoors, what options do I have? |
|---|---|---|
| 16 | https://www.zhihu.com/question/634 64857/answer/499357224 | How do you view the ban on DJI drones by US Military? |
| 17 | https://www.zhihu.com/question/267 112932/answer/515469926 | Why have quadcopters become popular, and helicopters not? |
| 18 | https://www.zhihu.com/question/263 726501/answer/272395138 | What's your opinion on the Indian drone that invaded China's airspace and crashed on 7 December? |
| 19 | https://www.zhihu.com/question/522 37274/answer/130230580 | What do new drone startup companies need to pay attention to when they want to make products like those of DJI? |
| 20 | https://www.zhihu.com/question/264 651961/answer/284223719 | How to perceive the practices of SF Groups to start using large drones for express delivery? |
| 21 | https://zhuanlan.zhihu.com/p/216160 26 | What is the killing power of drone rotors? |
| 22 | https://zhuanlan.zhihu.com/p/347639 73 | It only takes one after-sales service to make you transform from a DJI fan to an anti-fan. |
| 23 | https://www.zhihu.com/question/244 53944/answer/151209621 | What are the experiences of working at DJI? |
| 24 | https://www.zhihu.com/question/268 059844/answer/336556440 | Is the rumor that Israel has purchased tens of thousands of DJI's UAVs true? What's your opinions on this? |
| 25 | https://www.zhihu.com/question/279 68317/answer/40938920 | Why are there so many quadcopter fans in the country attacking DJI? |
| 26 | https://zhuanlan.zhihu.com/p/266638 50 | Strange unlicensed drone "black flights" near Chengdu Shuangliu Airport inspire many conspiracy theories. |
| 27 | https://zhuanlan.zhihu.com/p/345995 00 | When will DJI be restricted to export? How does a flying camera transform into a flying grenade? |
| 28 | https://www.zhihu.com/question/612 22947/answer/214464192 | What's your opinion on DJI passively being labeled as a military company? |
| 29 | https://www.zhihu.com/question/236 76158/answer/26514753 | What is the current market status of small civil drones and their future prospects? |
| 30 | https://www.zhihu.com/question/304 93677/answer/83639547 | A drone is always flying around in front of my window; how to deal with it? |
| 31 | https://www.zhihu.com/question/522 37227/answer/130450498 | Where does one need to pay attention to while traveling with a drone for aerial photography? |
| 32 | https://www.zhihu.com/question/522 37086/answer/129815145 | Is drone operation difficult? |
| 33 | https://www.zhihu.com/question/275 543118/answer/381574288 | What was the cause of the mistakes in the 1374 UAV show performances in Xi'an on 1 May 2018? |
| 34 | https://www.zhihu.com/question/382 45247/answer/171462696 | What beautiful pictures or videos have you taken by aerial photography? |
| 35 | https://www.zhihu.com/question/235 17562/answer/27359260 | What is the value of the class Signals and Systems? |
| 36 | https://zhuanlan.zhihu.com/p/324697 51 | 2017 Best Drone Photos |
| 37 | https://www.zhihu.com/question/679 09369/answer/261269277 | What interesting things have you encountered by aerial photography? |
| 38 | https://zhuanlan.zhihu.com/p/265916 12 | Under the great reward of Dajiang, there must be some conspiracy. |
| 39 | https://www.zhihu.com/question/275 869496/answer/393552974 | It is said that unlicensed drone 'black flight' alarmed the Air Force fighters and 4 people were arrested. What's your opinion on this? |
| 40 | https://www.zhihu.com/question/578 35306/answer/174110295 | How to evaluate DJI's small drone model Spark? |
| 41 | https://zhuanlan.zhihu.com/p/480459 84 | Northeastern Tigers jointly attacking drones |

| 42 | https://www.zhihu.com/question/273 45010/answer/88840753 | What technical problems exist with using drones for express delivery? |
|----|---|---|
| 43 | https://www.zhihu.com/question/511 26425/answer/124243444 | Is DJI famous abroad? How do foreigners evaluate it? |
| 44 | https://www.zhihu.com/question/280 76833/answer/39311347 | Why have drones become so popular? |
| 45 | https://www.zhihu.com/question/409 27288/answer/89323103 | How to evaluate DJI's drone model Phantom 4? |
| 46 | https://www.zhihu.com/question/278 057281/answer/397826100 | How do you view that a student of China University of Geosciences (Wuhan) has navigated a drone to hover around the girls' dormitory? Does this infringe on the privacy of others? |
| 47 | https://www.zhihu.com/question/268 998829/answer/352011397 | How to evaluate the variety show "Superpower Science"? |
| 48 | https://www.zhihu.com/question/279 04916/answer/39093164 | How is the modern advanced control theory advanced? How do you view the situation where the advanced control system is actually not that stable even though the PID controller has almost 90% share? |
| 49 | https://zhuanlan.zhihu.com/p/272483 55 | Half insect, half machine, live dragonfly-drone successfully tested, opening the "Cyborg" semi-mechanized biological era. |
| 50 | https://www.zhihu.com/question/239 71601/answer/26254459 | How to explain the Kalman filter as much as possible and as detailed as possible? |
| 51 | https://zhuanlan.zhihu.com/p/396065 71 | The hacker sold the U.S. military drone design files for $200 on the dark web. |
| 52 | https://www.zhihu.com/question/517 40470/answer/127847340 | What advantages does DJI have in its victory over 3D Robotics? |
| 53 | https://www.zhihu.com/question/215 84809/answer/18711695 | How do you view SF Express's drone delivery service that is being internally tested? |
| 54 | https://www.zhihu.com/question/333 20226/answer/56316028 | Hoping to build a robot myself, what professional knowledge should I know? |
| 55 | https://www.zhihu.com/question/599 13410/answer/170526318 | How do you view the Civil Aviation Bureau's implementation of the real-name registration system for civil drones since 1 June? |
| 56 | https://www.zhihu.com/question/291 486522/answer/476352652 | How to evaluate DJI Mavic 2 Pro and Mavic 2 Zoom? |
| 57 | https://zhuanlan.zhihu.com/p/368615 35 | The foreign brother bet he can let IKEA's chair go to heaven! Can it fly? After a few hours. |
| 58 | https://www.zhihu.com/question/265 27249/answer/33940148 | What exactly is DJI Inspire? |
| 59 | https://zhuanlan.zhihu.com/p/383216 61 | Beijing brother summoned the machine dragon in Japan and won the ICRA 2018 best drone paper |
| 60 | https://www.zhihu.com/question/597 65611/answer/168971569 | What's your opinion on the drone interference that Chongqing Airport encountered once again on the night of 12 May 2017? |

[1] Ranked by highest number of endorsements of the most popular answer (from top). Questions translated by the authors of this article.

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
