# Peer review of "“Why Drones for Ordinary People?” Digital Representations, Topic Clusters, and Techno-Nationalization of Drones on Zhihu"

_information, doi:10.3390/info10080256_

Round 1

Reviewer 1 Report

Review Report

Manuscript ID: information-554920.Type of manuscript: Article. Title :“Why Drones for Ordinary People?” Digital Representations, Topic Clusters, and Techno-Nationalization of Drones on Zhihu. Authors: Andrea Hamm *, Zihao Lin. Conference: Digital Citizenship and Participation 2018.

https://www.mdpi.com/journal/information/special_issues/digital_citizenship2018.

Brief summary.

The paper bring a mixed methods approach, Social Network Analysis (SNA) and Critical Discourse Analysis(CDA) to show that drones coupled with the Zhihu digital platform, (China’s largest question/answering platform) are breaking and creating new spaces for «image based communication» for ordinary well educated netizens. In their Social Network Analysis, Hamm and Lin discovered that the strong linkages of the drone topic to DJI and Aerial Photography imply a prevailing consumer perspective on drones as tools for aerial photography mediated by the public relations of the largest drone producer DJI. But what is an interesting output of their CDA, is that, it however revealed that such consumerist view on the drone technology is only one among several representations of drones: 1-as an object of consumption and technological advancement; 2-as an object of uncertainty; 3-as an object of national future

Hamm and Lin’s study demonstrate that drones are actively appropriated by users via national lens. Several answers and comments in their material corpus show a reference to the Chinese national development, global market leadership, or even weapon technology advancement. At the same time, the SQA platform engenders a space of attitudes to its users who are contributing in an intimate way to the online community. Drones open a discursive space for socially engaged people to make sense of the social changes that are unfolding. The changes encompass the emerging digital economies, the 9-9-6 work culture, new market regulation, consumerism, The China Dream, and technological breakthroughs.

General comments: A)The emergence of China as the largest global consumer drone producer and the advent of social media platform like the giant ZHIHU are two events that disrupt the world as it is and force a rethinking of social communication and digital public spaces. Social interaction in China, often performed on a «mediascape» dominated by social media under authoritarian state policy and large business organization like DJI, suggest new landscapes of possibility for netizens dialogs and a need to adopt new methods and tools for understanding the myriad practices of ordinary people that exceed the structures of governmental control and global business marketing. This research shows many controversial issues emerging practices of citizenship and new representations of technology in digital public spaces by coupling the observation of unmanned Aerial Vehicle and and Q/A platform Zhihu. However, despite a good conceptual framework, it seems to me that your research could make more sound reference with fields like Computer Mediated Communication (CMC), or Human Computer Interaction (HCI), or even Human Robot interaction?(HRI) Or also Community Informatics and Social Computing. Is it possible to add a few references to one or more of these fields to improve your conceptual framework?

B-Recently, drones (or unmanned aerial vehicles (UAVs) have been getting increasing attention and taking platform computing to a new era. Due to the support of highly advanced technologies, soon they might be ubiquitous and networks of drones might be used in providing civilian drone services. In your  paper, you provide a very good context for your research. You give a  survey of drone services and applications, where sense-making process emerge through conversations and narratives that can potentially have political effect.

These narratives trends are fuelled by drones data and a range of human-drone interaction research which is useful if drones are to regularly serve ordinary people. Could it be possible to give some more information about the political aspect of this social phenomenon? (emergence of democratic dialog versus an authoritarian conversation under external rules)?

C- Since 20 years Online Communities of interest are a rapidly growing phenomenon and are becoming increasingly important to society in the future. In China, this growth has been mirrored by scientific innovations: major advances have been made with regard to technology, software development, Knowledge Infrastructure, and now, drones and democratic platform conversations. However, we often overlook information on the ties that bind and that gives rise to virtual communities. While we are provided with an increasingly detailed topology of a network this does not allude to what content is at stake. So, in your paper you therefore propose to combine Social Network Analysis (SNA) and Discourse Analysis (DA) in order to reach a deeper understanding of the community. Data from an ongoing study of massive online citizen community is analyzed as an example using SNA and CDA. You present findings from SNA and that way you are able to complement this relational information with interpretive findings from CDA. In doing so, you substantially contribute to the methodological literature on online communities, in particular in the fields of SNA and DA. But unfortunately, your sections «discussion» and your implicit «conclusion» (last paragr. of the article) could be more explicit on that methodological subject. The combination of results from social network analysis and discourse analysis yields insights into the topology as well as the content of an online community. Social network analysis provides data about the network and the relations among its actors, whereas discourse analysis allocates meaning and sense-making within the network. By combining these two methods, do you want to bridge a technical gap? What are the advantages and possible disadvantages of your approach in term of qualitative and quantitative results? Moreover, when combining the interpretive qualities of discourse analysis with more descriptive content analysis techniques, insights might be gained regarding the semantic network within the whole Zhihu community and its sub-social systems? Could you provide us please with more information especially in the last section of your article, to better validate your effort and contribution?.

Specific comments for rating the manuscript

1-Originality; To my knowledge, this is the first study on UAV on Zhihu linking SNA and CDA in the analysis of the 60 most popular questions on this citizen platform.

2-Significance: The results are appropriately interpreted. However, the conclusion must be appearing after the discussion to correspond to MDPI standards. More specifically, a conclusion should be drawn from the epistemological performance of the combination of SNA and CDA.

3-Quality of presentation: The paper is structures in the following way: Abstract, Introduction, Context is well developed, Conceptual Framework (a little lack of reference to other important fields like Human Computer Interaction, or Computer Mediated Communication), Research Design, (very explicit), Findings (very well developed), Discussion. It could be interesting to add a Conclusion around my previous remarks on the SNA et CDA methodologies, and the future research it gives rise. A conclusion that must be more specific and developed around this new context of research and new opportunities for mixed methods in area such as Community Informatics.

4-Scientific Soundness: The research is correctly designed and technically sound. The analysis are performed with high technical standards. The data are enough to draw a better Conclusion on the whole methodological process. The methods, the tools, the platforms are well detailed in «context» section. I think that I could use it myself to conduct a similar study.

5-Interest to the readers: I think that the paper present a «new side» and a new ideological vision  of the digital representations of Chinese netizens that is not the usual western vision that describe the lack of democracy in China without appreciating the real link between people on these new platforms. So many studies are denunciating the Social Credit System in China, without the sufficient knowledge of the political history in China.

6-Overall Merit: The work provide an advance towards the current ideological western research that says that in general, there is no democracy in China and that its citizen have absolutely no right of expression. A methodological innovation, but not explicit enough in the discussion.

7-English level: English language is appropriate and minor revisions are required. Pay attention please on little spelling mistake like in section 3; where you wrote «Conceptional Framework» instead of conceptual framework.

8-Tables, photos and figures : In my black version of the paper, photos and figures are not clear enough for the readers. Also, in figure 6 page 12, what are the real meanings or the role of these photos for the understanding of the whole paper? Just a few precisions please.          

Overall recommendation:

Accept after minor revisions 

Commentary to the authors: Future research should take into account the advantages that a combination of methods yields. There are several directions that are as yet unexplored, but promise fruitful results. One possibility is to repeat the analysis you performed in this article several times. This way, results could be interpreted over time and thus reveal network evolution on Zhihu, both in a structural as well as discursive and interpretative way. Second, it might be interesting to compare the discursive profile of several actors. For example, do profiles of central actors differ significantly from the ones of peripheral actors? Do

central actors feature similar profiles? Does a profile predict network position in the future? These are all compelling questions, and answers could help us understand China’s online communities representations much better than we do now.

Kind regards,

Pierre-Léonard Harvey, Ph.D

Author Response

Response to general comment A: We added the key references from the field of computer-mediated communication and virtual community informatics in the conceptual framework, and correspondingly in the conclusion, discussion.

Response to general comment B: In the conceptual framework, we made more explicit the political implications of drone technologies by distinguishing a utopian and dystopian approach to drones in existing literature- and we suggested to go beyond this dichotomic logic by conceptualizing drone as a mediator of social change in Latourian sense.

Response to general comment C: We separated the section of Discussion and Conclusion while including reflections on the methodological contributions of CDA+SNA.

Response to English Level: We conducted a thorough recheck of the grammatical composition of the whole manuscript.

Response to Tables, Photos, and Figures: To achieve better readability, we deleted the photo screenshots in Findings section and instead incorporated necessary analysis of visual materials in the result if there haven't been such.

Reviewer 2 Report

The paper is well-thought. The research design and methods are sufficiently and clearly discussed in the paper. I just have a minor suggestion which I may have missed out while reading your paper, but could be taken into consideration as you finalise your paper for publication:Why are some sentences, phrases, words bold or underlined? Is it possible for you to insert a line or two to explain the reason why this is so? Thank you. 

Author Response

Response to comment: We added a short explanation of text formatting in the end of section 4. In addition, We conducted a thorough recheck of the grammatical and lexical composition of the whole manuscript.

Round 2

Reviewer 1 Report

in conclusion and from my side, this latest manuscript has been significantly improved and now warrants publication in information.